

# Seasonal dynamics alter taxonomical and functional microbial profiles in Pampa biome soils under natural grasslands

Anthony Diego Muller Barboza[1], Victor Satler Pylro[2],
Rodrigo Josemar Seminot Jacques[3], Paulo Ivonir Gubiani[3],
Fernando Luiz Ferreira de Quadros[4], Júlio Kuhn da Trindade[5],
Eric W. Triplett[6] and Luiz Roesch[1]

[1] Centro Interdisciplinar de Pesquisas em Biotecnologia—CIP-Biotec, Universidade Federal do Pampa, São Gabriel, Brazil
[2] Department of Soil Science, "Luiz de Queiroz" College of Agriculture, University of São Paulo—ESALQ/USP, Piracicaba, Brazil
[3] Departamento de Solos, Programa de Pós-graduação em Ciência do Solo, Universidade Federal de Santa Maria, Santa Maria, Brazil
[4] Department of Animal Science, Universidade Federal de Santa Maria, Santa Maria, Rio Grande do Sul, Brazil
[5] Departamento de Diagnóstico e Pesquisa Agropecuária, Secretaria Estadual da Agricultura, Pecuária e Irrigação, São Gabriel, Brazil
[6] Department of Microbiology and Cell Science, University of Florida, Gainesville, United States of America

Corresponding author
Luiz Roesch,
luizroesch@unipampa.edu.br

## ABSTRACT

Soil microbial communities' assembly is strongly tied to changes in temperature and moisture. Although microbial functional redundancy seems to overcome taxonomical composition changes, the sensitivity and resilience of soil microbial communities from subtropical regions in response to seasonal variations are still poorly understood. Thus, the development of new strategies for biodiversity conservation and sustainable management require a complete understanding of the soil abiotic process involved in the selection of microbial taxa and functions. In this work, we used state of the art molecular methodologies (Next Generation Sequencing) to compare the taxonomic (metataxonomics) and functional (metatranscriptomics) profiles among soil samples from two subtropical natural grasslands located in the Pampa biome, Brazil, in response to short-term seasonal variations. Our data suggest that grasslands maintained a stable microbial community membership along the year with oscillation in abundance. Apparently soil microbial taxa are more susceptible to natural climatic disturbances while functions are more stable and change with less intensity along the year. Finally, our data allow us to conclude that the most abundant microbial groups and functions were shared between seasons and locations reflecting the existence of a stable taxonomical and functional core microbiota.

## INTRODUCTION

The selection imposed by abiotic environmental factors is an important event contributing to microbial community assembly (*Stegen et al., 2012*). Although soil microbial

communities appeared to be well adapted to some environmental variability (e.g., increase of 4 °C in soil temperatures) (*Schindlbacher et al., 2011*), short-term seasonal changes are strongly correlated with shifts in microbial community composition (*Wallenstein & Hall, 2012*). Diversity and/or abundance of microbial taxa changed between seasons in alpine ecosystems, (*Lipson & Schmidt, 2004*), temperate grassland ecosystem (*Habekost et al., 2008*), soils of a temperate beech forest (*Rasche et al., 2011*) and soils under Mediterranean climate (*Waldrop & Firestone, 2006*).

Among the deterministic processes governing the composition of microbial communities, environmental factors such as soil pH, temperature and moisture are considered to be main drivers of microbial community assembly (*Castro et al., 2010*; *Fierer & Jackson, 2006*; *Kaiser et al., 2016*; *Lauber et al., 2013*). Particularly, differences in beta diversity among soils are significantly correlated with soil pH (*Lauber et al., 2009*). In a large sampling study including 150 forest and 150 grassland soils *Kaiser et al. (2016)* found soil pH as the best predictor for bacterial community structure, diversity and function. Under normal rainfall conditions the increase in temperature positively affects the abundance of some soil microbial groups (*Sheik et al., 2011*). Conversely, increasing temperature was positively correlated with microbial respiration in soil and $CO_2$ production (*Luo et al., 2001*), which in turn can reduce the population of *Actinobacteria* sensitive to high $CO_2$ concentrations, for example (*Sheik et al., 2011*). Water saturation can drastically alters soil microbial composition, community diversity and function by altering $O_2$ availability (*Carbone et al., 2011*). On the other hand, drought conditions may benefit organisms adapted to arid conditions (*Bouskill et al., 2013*).

In subtropical regions, soil temperature and moisture vary strongly during the seasons, influencing microbial community assembly. However, previous work suggests that soil functioning is not affected by the decline of the microbial diversity (*Griffiths et al., 2001*; *Wertz et al., 2007*; *Wertz et al., 2006*), and these ecosystems seems to rely on functional redundancy (*Frossard et al., 2012*; *Lupatini et al., 2013*; *Nannipieri et al., 2003*; *Rousk, Brookes & Baath, 2009*).

Although microbial composition changes are not necessarily correlated with soil microbial functions, the sensitivity and resilience of soil microbial communities from subtropical regions in response to seasonal variations are still unclear. Obtaining a better insight into the soil ecological process involved in the selection of microbial taxa and function is paramount for developing new strategies for biodiversity conservation and sustainable management. Moreover, if microbial taxa shift along the year and functions do not change, such information is relevant for planning sampling strategies and for comparing the results from different experiments. Here, we compared taxonomic and functional microbial profiles of soils under subtropical natural grassland from two sites in the Pampa biome, during the warm (summer and autumn) and cold (winter and spring) seasons. The major aim of this study was to test whether short-term seasonal variations (mainly temperature and precipitation), typical in subtropical regions, can cause changes in microbial phylotypes and/or microbial functions. We expected microbial community abundance and diversity patterns to change due seasonal variation, that is, to increase over the warm season, but microbial functioning to remain stable over seasons.

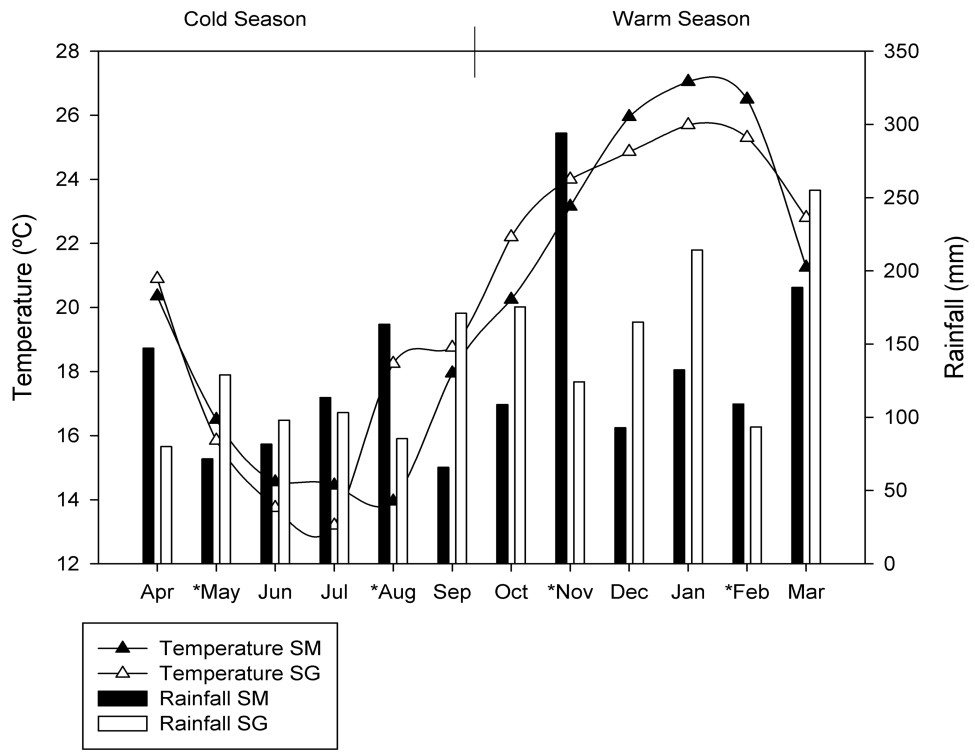

**Figure 1** **Seasonal dynamics of temperature and rainfall in two natural grasslands (SG and SM) from the Pampa biome.** The asterisks depict the sampling period.

# MATERIAL AND METHODS

## Study sites

The study was conducted during 2014 and 2015 in two sites under similar environmental conditions located in the Brazilian Pampa biome: Santa Maria municipality—SM (29°45′S, 53°45′W) and São Gabriel municipality—SG (30°20′S, 54°15′W), 83 km far from each other. The sites were chosen because of their similarities in soil features, vegetation, land use, and climate. Soil samples were collected under natural grassland, currently used for grazing of cattle, with no historic of fertilizers input (except for the manure added by animal activity). The sites exhibited similar soil physicochemical characteristics (Table S1) and were classified within the same soil taxonomy.

The sites chosen are exposed to large seasonal variations in soil moisture and temperature (Fig. 1). For this study, the seasons were characterized as warm season (October, November, December, January, February and March) and cold season (April, May, June, July, August and September), due to the similarities in temperature and rainfall between spring and summer as well as autumn and winter. The average air temperatures in the cold season were 16.2 °C ± 2.4 and 17.5 ± 2.7 °C in SM and SG, respectively. In the warm season the average air temperatures were 24.0 ± 2.8 °C in SM and 24.1 ± 1.4 °C in SG. The monthly average rainfall in the cold season was 107 mm in SM and 112 mm in SG. In the warm

season, monthly average rainfall was 154 mm and 171 mm, in SM and SG, respectively ([Fig. 1]).

The plant species comprising 95% of the total biomass in SM were: *Adropogon lateralis, Aristida laevis, Elephantopus mollis, Bacharis trimera, Paspalum plicatulum* and *Paspalum notatum,* and in SG were: *Andropogon lateralis, Axonopus affinis, Baccharis trimera, Eragrostis plana* (invasive), *Erianthus angustifolius, Paspalum notatum, Paspalum dilatatum, Paspalum plicatulum, Sporobolus indicus* and *Vernonia nudiflora.*

## Soil sampling, nucleic acids extraction and preprocessing

At each grassland (SM and SG), the sampling area was divided in three blocks, and five soil cores per block were randomly selected and pooled to make a single composite sample, resulting in three samples per site at each season ($n = 24$, that is, two sites (SM and SG), 4 months (May and August, 2014—cold season/November 2014 and February 2015—warm season) and three replicates per sampling). Samples were collected with a sterile spatula to a depth of 5 cm, kept in sterile 50 mL tubes in liquid nitrogen, and then stored at $-80\ °C$ until DNA and RNA co-extraction.

From each sample, 2 g of bulk soil (free of roots or rhizospheric soil) was used for simultaneous extraction of total RNA and DNA using the RNA PowerSoil kit and the PowerSoil® DNA Elution Accessory Kit (MoBio laboratories, Inc., Carlsbad, CA, USA), following the manufacturer's instructions. After RNA extraction, each sample was subjected to enzymatic digestion of DNA with TURBO DNA-free Kit (Applied Biosystems, Foster City, CA, USA). All RNA and DNA samples were submitted to quantification and quality-check using a Qubit RNA or DNA assay kit (Invitrogen, Eugene, OR, USA) and NanoVue™ spectrophotometer (GE healthcare, Chicago, CA USA), respectively, and further stored at $-80\ °C$ until library preparation.

## Metatranscriptome library preparation and sequencing

One microgram of total RNA from each bulk soil sample were depleted by removing rRNA from total RNA with the MICROBExpress™ Bacterial mRNA Enrichment Kit (Thermo Fisher, Waltham, MA, USA) following the manufacturer's instructions. The mRNA was further purified with the MEGAclear™ Transcription Clean-Up Kit (Thermo Fisher, Waltham, MA, USA) following the manufacturer's instructions. The enriched mRNA was used to prepare the mRNA library with the Ion Total RNA-Seq Kit v2 and Ion Xpress™ RNA-Seq Barcode Kit (Thermo Fisher, Waltham, MA, USA). The libraries were amplified by emulsion PCR and sequenced on Ion 316™ Chip v2 using the Ion Torrent PGM system and the Ion PGM™ Sequencing 400 kit, according to the supplier's instructions. After sequencing, all reads were filtered by the PGM software to remove low quality and polyclonal sequences.

## 16S library preparation and sequencing

Library preparation and sequencing followed the procedures described by *Suleiman et al. (2016)*. For analyzing the composition of soil bacterial and archaeal communities the total microbial DNA was used as template for partial 16S rDNA amplification and sequencing. The V4 region of the 16S rRNA gene was amplified using the bacterial/archaeal primers 515F

and 806R (*Caporaso et al., 2012a*), and sequenced using the PGM Ion Torrent (Thermo Fisher Scientific, Waltham, MA, USA). Multiple samples were PCR-amplified using barcoded primers. Each of the 25 μL of PCR mixture consisted of 2U of Platinum® Taq DNA High Fidelity Polymerase (Invitrogen, Carlsbad, CA, USA), 4 μL 10× High Fidelity PCR Buffer, 2 mM MgSO4, 0.2 mM dNTP's, 0.1 μM of both the 806R barcoded primer and the 515F primer, and approximately 100 ng of DNA template. The PCR conditions used were 95 °C for 5 min, 30 cycles of 94 °C per 45s denaturation; 56 °C per 45 s annealing and 72 °C per 1 min extension; followed by 72 °C per 10 min. The resulting PCR products were purified with the Agencourt® AMPure® XP Reagent (Beckman Coulter, Brea, CA, USA) and the final concentration of the PCR product was quantified by using the Qubit Fluorometer kit (Invitrogen, Carlsbad, CA, USA) following the manufacturer's recommendations. Finally, the reactions were combined in equimolar concentrations to create a mixture composed by 16S rRNA gene amplified fragments of each sample. This composite sample was used for library preparation with Ion OneTouch™ 2 System with the Ion PGM™ Template OT2 400 Kit Template (Thermo Fisher Scientific, Waltham, MA, USA). The sequencing was performed using Ion PGM™ Sequencing 400 on Ion PGM™ System using Ion 316™ Chip v2.

## Bioinformatics analysis and statistics

The annotation of metatranscriptome sequences was performed with the Metagenomics Rapid Annotation (MG-RAST) pipeline version 3.3.6 (*Meyer et al., 2008*), using the standard parameters for sequence quality control. The data was compared to the SEED Subsystem using a maximum $e$-value of $1^{-5}$, a minimum identity of 60%, and a minimum alignment length of 15 measured in aa for protein and bp for RNA databases. The 16S rRNA raw sequences were analyzed following the recommendations of the Brazilian Microbiome Project (*Pylro et al., 2014*), using the BMP Operating System (BMPOS) (*Pylro et al., 2016*).

Analysis of Metagenomic Profiles v2 (STAMP) software package was used to determine differences in relative abundances of microbial functions and taxa (*Parks et al., 2014*). Statistical hypothesis tests were performed using the Welch's $t$ test while confidence intervals were calculated using the Welch's inverted method and Bonferroni multiple test for $p$-value correction. The mRNA and the 16S rRNA datasets were rarefied to the same number of sequences per database (*Lemos et al., 2011*) and used to construct dissimilarity matrixes generated by Bray Curtis and Binary distances using the "phyloseq" package in R. The matrixes were ordinate by Principal Coordinate Analysis (PCoA) and *adonis* function was used to calculate the Permutational Multivariate Analysis of Variance (PERMANOVA) and verify the strength and statistical significance of groups among location, season and the combined effect of location and season with the vegan package (*Oksanen et al., 2015*). Sampling effort was estimated using Good's coverage (*Good, 1953*).

Raw sequences obtained by metatranscriptome sequencing and associated metadata were submitted to MG-RAST server (http://metagenomics.anl.gov/) and are publicly available under the string mgp12046. All raw sequences obtained by amplicon sequencing were submitted to NCBI Sequence Read Archive (SRA) and are available under the experiment number SRX2779549 and run number SRR5499445.

## RESULTS

### Overall microbial differences among seasons and locations

After quality filtering the 16S rRNA reads, a total of 1,630,136 high-quality sequences longer than 200 bp were retained. The average Good's coverage of 98% was calculated (Table S2), indicating the dataset was representative of the microbial communities analyzed. The soil metatranscriptome sequencing yielded a total of 20,584,533 reads for all 24 samples, and the average Good's coverage was 77% (Table S2).

To evaluate the differences in microbial community structure and function between the two natural grasslands and along different seasons, the abundances of microbial taxa and mRNA encoding functions were used to compute a Bray–Curtis dissimilarity matrix coordinated by using PCoA. A Binary similarity matrix was also used to compute differences of either presence/absence of taxa or mRNA encoding functions. Contrasting patterns of microbial taxonomic structure and functions were observed. Although the sites were similar, these analyses revealed differences in microbial taxonomic structure between the two natural grasslands (SM and SG) and between seasons (Fig. 2). Four distinctive groups were observed taking into account the microbial taxonomic structure meaning the soil microbial phylotypes shifted along the year according to the season. When the abundance of mRNA encoding microbial functions was compared, two distinctive groups were observed, representing the cold and the warm seasons. Those results indicate that microbial functions are more resilient to environmental changes than the microbial taxonomic structure. No significant differences in microbial taxonomic structure or microbial functions were observed within the same season in either location. The location did not influence the main functions performed by the microbial communities (Fig. 2). Microbial community differences were further confirmed by PERMANOVA (Table 1). Season was the main driver of shifts of abundance of taxa and mRNA encoding functions of the soil microbiota. This factor contributed 22% and 23% for the variation in microbial taxa and mRNA encoding functions, respectively (Table 1). Altogether, these results indicated that soil microbial community structure was affected by location (e.g., two natural grasslands from the same biome, but 83 km apart) and by the natural environmental changes throughout the year (seasons). Also, location and season interaction was significant at 5% probability, meaning the impact of one factor depends on the level of the other factor or, in other words, that microbial community shifts emerge from the interaction location and seasons. On the other hand, the functions performed by soil microbes in those two sampling sites were similar, not being affected by location, but by seasons.

### Identification of taxa and functions that shifted along seasons and locations

Once overall taxonomic and functional differences were identified, the functional differences as measured through RNAseq, and taxonomical differences as measured by 16S rRNA sequencing, were determined. The seasonal frequency of most abundant microbial phylotypes (relative frequency greater than 1%) in both locations was determined (Table 2). The data indicated the grassland ecosystem maintained a core microbiota along the year but the relative abundance of taxa varied in response to seasons. As the amount of rain

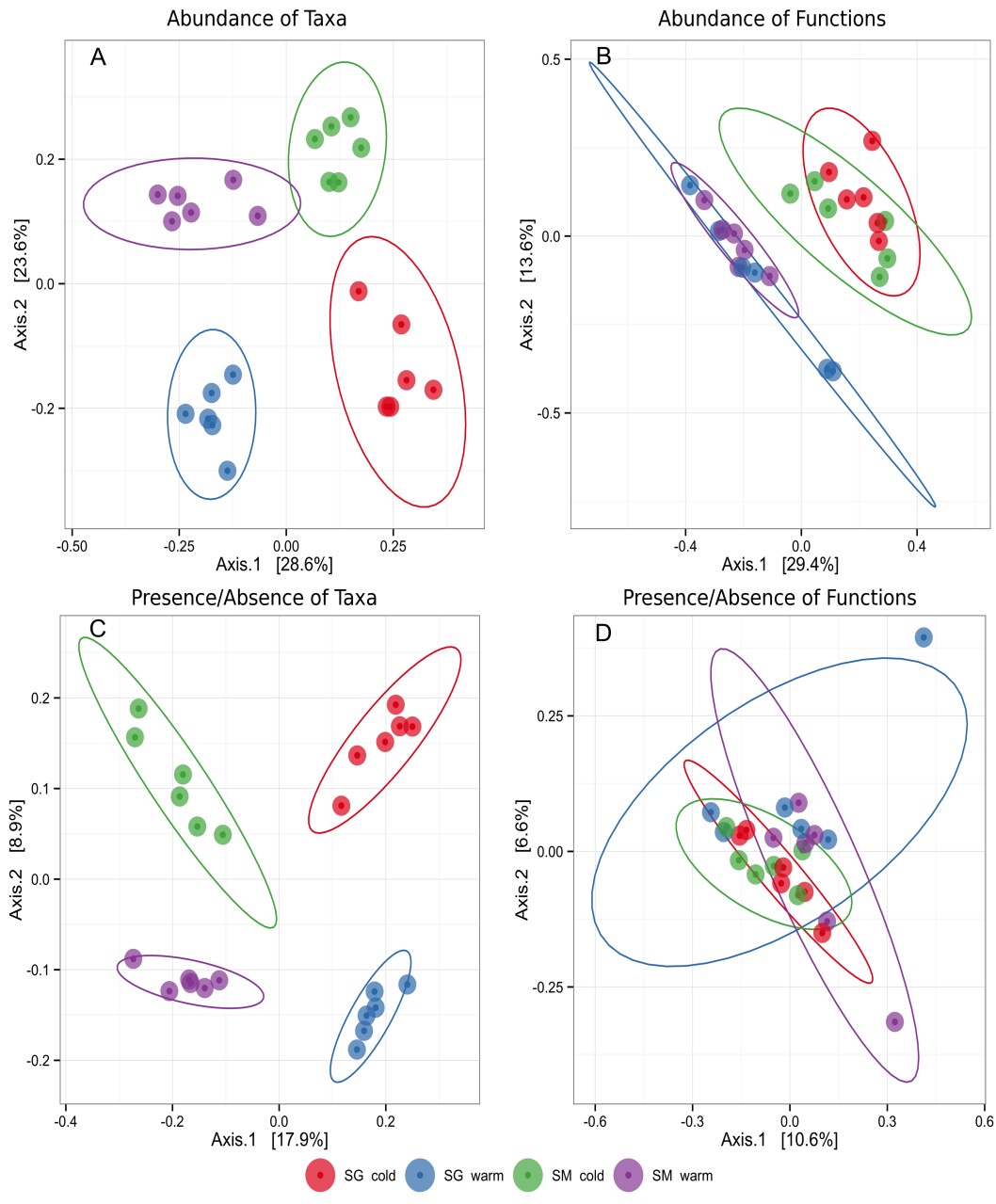

**Figure 2** **Principal coordinates plot (PCoA) representing clusters of soil microbial communities grouped by taxa or functions in two different natural grasslands (SG and SM) and seasons (cold and warm) in the Pampa biome.** Ellipses around groups represent 95% confidence intervals for the standard error of the average scores for each group. (A) Abundance of microbial taxa. (B) Abundance of microbial functions. (C) Presence/absence of microbial taxa. (D) Presence/absense of microbial functions.

was well distributed along the year (Fig. 1), soil moisture was not considered an important variable shaping microbial community assembly in our experiment. Temperature shifted during the year with an average minimum of 10 °C and an average maximum of 27 °C, representing an important source of environmental variation influencing microbial life

**Table 1** Multivariate analysis of variance showing the differences between soil microbial communities and functions.

| Factors | Abundance of taxons | | Abundance of mRNA encoding functions | |
| --- | --- | --- | --- | --- |
| | $R^2$ | $p$-value | $R^2$ | $p$-value |
| Location | 0.262 | 0.001 | 0.036 | 0.327 |
| Season | 0.223 | 0.001 | 0.234 | 0.001 |
| Local × Season | 0.055 | 0.016 | 0.042 | 0.214 |

| | Presence/absence of taxons | | Presence/absence of mRNA encoding functions | |
| --- | --- | --- | --- | --- |
| | $R^2$ | $p$-value | $R^2$ | $p$-value |
| Location | 0.085 | 0.002 | 0.040 | 0.696 |
| Season | 0.170 | 0.001 | 0.054 | 0.019 |
| Local × Season | 0.051 | 0.043 | 0.042 | 0.427 |

**Table 2** Seasonal frequency of major bacterial groups detected in soil samples collected in two natural grasslands during different seasons.

| Phylum/genus[a] | Location SG | | |
| --- | --- | --- | --- |
| | Warm season | Cold season | |
| | Relative frequency ± std. dev. (%) | | $p$-values |
| *Acidobacteria/Candidatus Koribacter* | 2.25 ± 0.73 | 1.20 ± 0.55 | 0.0293 |
| *Actinobacteria/Mycobacterium* | 0.16 ± 0.04 | 1.31 ± 0.60 | 0.0080 |
| *Firmicutes/Bacillus* | 0.54 ± 0.51 | 2.46 ± 1.45 | 0.0301 |
| *Proteobacteria/Rhodoplanes* | 2.89 ± 0.37 | 6.26 ± 1.05 | 0.0004 |
| *Verrucomicrobia/Candidatus Xiphinematobacter* | 0.98 ± 0.59 | 3.17 ± 1.65 | 0.0298 |
| *Verrucomicrobia/DA101* | 23.76 ± 7.57 | 7.79 ± 4.05 | 0.0035 |

| | Location SM | | |
| --- | --- | --- | --- |
| | Warm season | Cold season | |
| | Relative frequency ± std. dev. (%) | | $p$-values |
| *Actinobacteria/Mycobacterium* | 0.28 ± 0.06 | 1.34 ± 0.48 | 0.0040 |
| *Proteobacteria/Rhodoplanes* | 1.83 ± 0.51 | 5.35 ± 1.23 | 0.0007 |
| *Verrucomicrobia/DA101* | 24.89 ± 11.89 | 5.64 ± 2.50 | 0.0143 |

**Notes.**

[a] The Welch's $t$-test was performed to obtain the $p$-values for the null hypothesis of no difference between warn and cold seasons. Only the genera with abundance greater than 1% are depicted here.

in our experiment. During the warm season, the uncultured member of *Verrucomicrobia* designated *DA101* was the most abundant phylotype in both locations. *Mycobacterium and Rhodoplanes* were also found in both grasslands with similarly high abundances, but better adapted to cold conditions (Table 2).

In agreement with the differential abundance analysis, alpha diversity measurements indicated similar microbial richness (number of observed OTUs) during the seasons, but greater microbial diversity (Simpson diversity index) during the cold season (Fig. 3). Cold
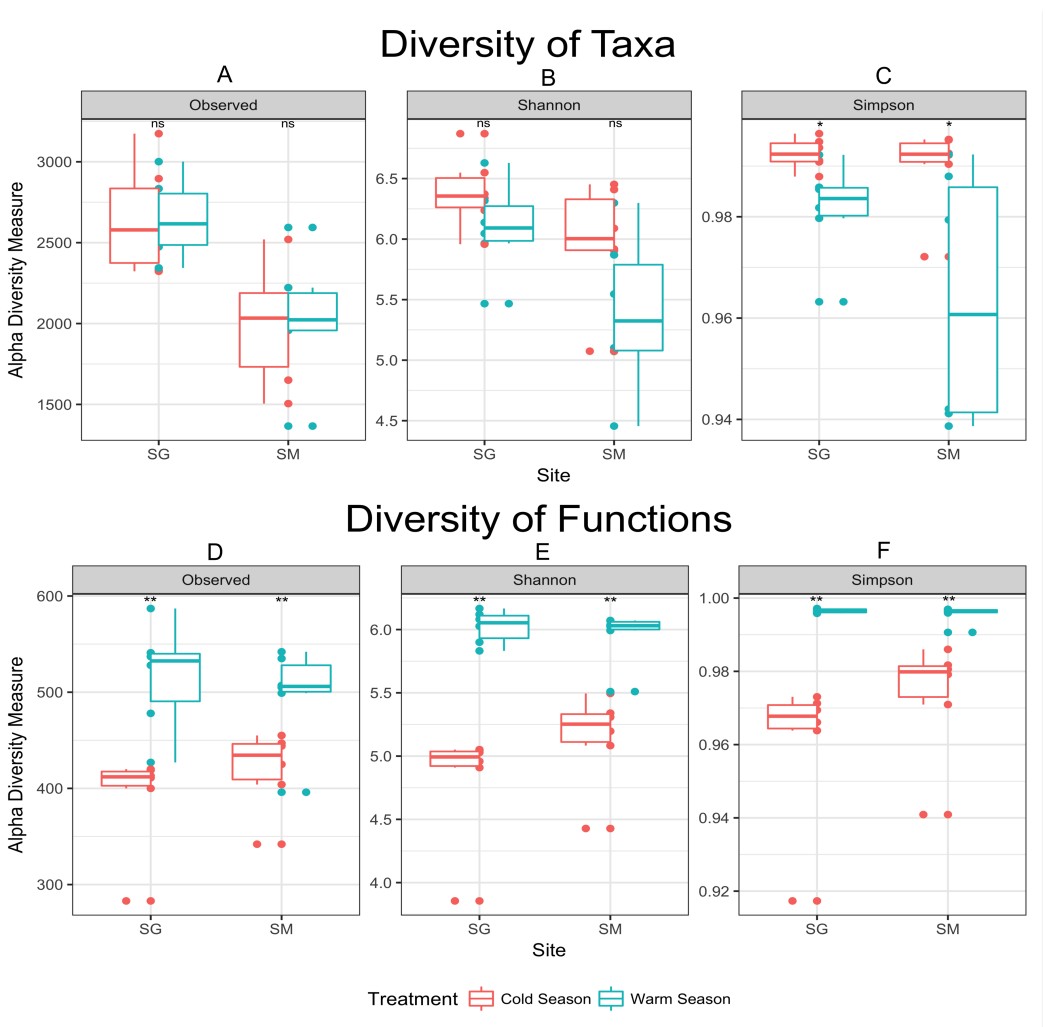

**Figure 3 Alpha diversity measurements of microbial taxa and functions during cold and warm seasons in two grasslands from the Pampa biome.** Each panel represents one alpha diversity measurement as follow: (A) total number of OTU's observed. (B) Shannon diversity index calculated for microbial taxa. (C) Simpson diversity index calculated for microbial taxa. (D) Total number of functions observed. (E) Shannon diversity index calculated for microbial functions. (F) Simpson diversity index calculated for microbial functions. Boxes span the first to third quartiles. The horizontal line inside the boxes represents the median. Whiskers extending vertically from the boxes indicate variability outside the upper and lower quartiles, and the single circles indicate outliers. Both datasets were rarefied to the same number of sequences before alpha diversity measurements. (ns) Non significant difference at $p$-value $\leq 0.05$ according to the pairwise $t$ test. (*) Significant difference at $p$-value $\leq 0.05$ according to the pairwise $t$ test. (**) Significant difference at $p$-value $\leq 0.01$ according to the pairwise $t$ test.

temperatures allowed for greater evenness while warm temperatures decrease the diversity of taxa. The diversity of microbial functions presented similar trends (high diversity during the warm season and low diversity during the cold season), displaying a core set of microbial functions along the year in both locations (Fig. 3 and Table 3). Main functional differences between seasons were consistently detected within both grasslands. Despite the difficulties

**Table 3   Relative abundance of mRNA encoding functions during cold and warm seasons in two different grasslands located in the Pampa biome.**

| SEED subsystems[a] | | | Location SG | |
|---|---|---|---|---|
| Level 1 | Level 2 | Level 3 | Cold season | Warm season |
| | | | Mean rel. freq. (%) ± SD | |
| Carbohydrates | Carbohydrates | Sugar utilization in Thermotogales | **10.5 ± 1.9** | 1.1 ± 0.3 |
| Carbohydrates | Monosaccharides | D-ribose utilization | **10.0 ± 2.0** | 0.4 ± 0.1 |
| Carbohydrates | Monosaccharides | Deoxyribose and deoxynucleoside catabolism | **10.0 ± 2.0** | 0.4 ± 0.1 |
| RNA metabolism | RNA metabolism | Group II intron-associated genes | **7.0 ± 2.5** | 2.0 ± 0.7 |
| Protein metabolism | Protein degradation | Proteolysis in bacteria, ATP-dependent | 1.2 ± 0.3 | **2.4 ± 0.5** |
| Carbohydrates | One-carbon metabolism | Serine-glyoxylate cycle | 1.0 ± 0.3 | **1.4 ± 0.3** |
| Protein metabolism | Protein biosynthesis | Universal GTPases | 0.9 ± 0.2 | **1.4 ± 0.2** |
| Cofactors, vitamins, prosthetic groups, pigments | Folate and pterines | YgfZ | 0.8 ± 0.1 | **1.2 ± 0.2** |
| Motility and chemotaxis | Flagellar motility in prokaryota | Flagellum | 0.5 ± 0.1 | **0.7 ± 0.1** |
| Protein metabolism | Protein degradation | Proteasome bacterial | 0.5 ± 0.1 | **0.8 ± 0.2** |
| RNA Metabolism | Transcription | Transcription initiation, bacterial sigma factors | 0.5 ± 0.1 | **1.0 ± 0.1** |
| Protein metabolism | Protein folding | Protein chaperones | 0.5 ± 0.2 | **1.1 ± 0.4** |
| | | | **Location SM** | |
| Carbohydrates | Carbohydrates | Sugar utilization in thermotogales | **7.1 ± 1.0** | 0.9 ± 0.2 |
| Carbohydrates | Monosaccharides | Deoxyribose and deoxynucleoside catabolism | **6.4 ± 0.9** | 0.2 ± 0.1 |
| Carbohydrates | Monosaccharides | D-ribose utilization | **6.4 ± 0.9** | 0.3 ± 0.1 |
| Protein metabolism | Protein degradation | Proteolysis in bacteria, ATP-dependent | 1.4 ± 0.4 | **2.3 ± 0.5** |
| Carbohydrates | One-carbon metabolism | Serine-glyoxylate cycle | 1.2 ± 0.1 | **1.5 ± 0.1** |
| Cofactors, vitamins, prosthetic groups, pigments | Folate and pterines | YgfZ | 0.9 ± 0.2 | **1.4 ± 0.2** |
| Protein metabolism | Protein biosynthesis | Universal GTPases | 0.8 ± 0.2 | **1.2 ± 0.2** |

**Notes.**

[a] The Welch's $t$-test was performed to obtain the $p$-values for the null hypothesis of no difference between warn and cold seasons. Only the functions with abundance greater than 1% and with difference between treatments with significant $p$-values ($\leq 0.05$) are depicted here. Numbers highlighted in bold represent greater abundance during either cold or warm season.

of defining the functions codified by complex mixtures of mRNA in metatranscriptome library, the metabolism of carbohydrates was the dominant set of functions performed during the cold season in both grasslands. An average of 10.5% of mRNA with known annotated function was committed to carbohydrate degradation and utilization in the model bacterium *Thermotoga maritima,* during the cold season in the grassland located in SG. Nevertheless, only 1.1% of the mRNA sequences were related with this function in the warm season (Table 3). The same pattern was observed in the grassland located in SM (7.1% of mRNA sequences committed to sugar utilization during the cold season and only 0.9% during the warm season). D-ribose utilization and deoxyribose and deoxynucleoside catabolism were also functions highly expressed during the cold season in both grasslands. During the warm season functions were more evenly distributed, without dominance of specific ones. The most abundant set of functions during the warm season were those

 

dedicated to the protein metabolism, composing 5.7% of mRNA dataset in the grassland located in SG, and 3.5% in the SM grassland. The functions related to Cofactors, Vitamins, Prosthetic Groups and Pigments were also higher expressed during the warm season in both grasslands.

## DISCUSSION

This study aimed to understand how and to what extent seasonal dynamics influence taxonomical and functional microbial profiles in subtropical natural grassland soils, based on metataxonomics (16S rRNA gene) and metatranscriptomics (mRNA) in-depth sequencing (as defined by *Marchesi & Ravel (2015)*). We addressed two important variables; season and location, aiming to better understand how these shifting environmental conditions influence microbial community assembly and function in subtropical grasslands from south Brazil. The Brazilian Pampa occupies 2% of the Brazilian territory and is considered one of the most fragile biomes in the country, experiencing losses of both biodiversity and socio-economic opportunities (*Roesch et al., 2009*). Natural grasslands are the predominant vegetal cover of this biome, although the climatic conditions are also suitable for forest development, which reflects its biological uniqueness. The soils in both sampling areas (SM and SG) were classified as Paleodult and the vegetal composition was similar but not identical. Our sampling strategy allowed us to obtain a highly tractable model (confounding variables like soil type and plant cover were controlled), with true landscape level of biological replications, for verifying whether soil microbial functions follow the changes in the abundance of taxa during seasonal variations. Our study revealed consistent effects of season on both microbial community structure and functions, with the former presenting less stability than the latter along the seasons.

Season and location significantly modulate microbial community assemblage and abundance in subtropical natural grasslands (Fig. 2). However, considering only those microbes with relative abundance greater than 1%, our data suggest that both grasslands maintained a stable microbial community membership along the time, but the relative abundance of specific taxa oscillated in response to seasonal changes (Table 2). Hence, these results support the existence of a soil core microbiota in natural grasslands that might represent the first step in defining a 'healthy' community and predicting community responses to perturbation (*Shade & Handelsman, 2012*). The concept of core microbiota was first introduced during the studies of the Human Microbiome Project (*Turnbaugh et al., 2007*). While the concept of microbial core remain elusive and might lack a conceptual framework involving microbial ecological characteristics (*Shade & Handelsman, 2012*), typical approaches report the presence/absence of microbial phylotypes across habitats. In the Pampa biome, *Lupatini et al. (2013)* already found a core microbiota among soil samples under Acacia plantation, natural pasture, soybean field, and natural forest. Within this study, 54.5% of OTUs (defined by 16S rRNA sequences with 97% similarity) were shared among four different land uses. Similarly, (*Suleiman et al., 2013*) found a total of 69% of the OTUs shared between a natural grassland and a pristine forest in the Brazilian Pampa biome. In our study, with the same land use, the presence of a soil microbial core

exists in which the most abundant bacterial groups were shared between different seasons and locations.

While a group of microbes were found to be able to survive throughout the year, relevant shifts in abundance of those groups along the seasons were apparent. These changes might be explained by nutrient fluxes caused by carbohydrates inputs (monosaccharides like glucose and fructose, oligosaccharides like maltose and sucrose and polysaccharides like cellulose and pectin) from senescent plants. Plant biomass production is low or even zero during the winter due to the climatic conditions that affect the most abundant plant species from natural grasslands. In south Brazil, the cold season covers around one third to half of the year. Low temperatures and frosts decrease the plant growth and plant senescence is intensified causing most of the forage to be rejected by the animals during the grazing. The entire process exacerbates the accumulation of litter and carbohydrates in soil. Under such conditions the availability of abundant and different carbon sources might be responsible for the changes observed in our experiment. Bacteria have access to greater availability of rhizodepositions in the summer but in the winter, when the rhizodeposition is very reduced, the easily decomposable carbon sources are very scarce in soil.

Cold conditions decreased the abundance of some taxa below the detection limit provided by our sequencing coverage. The apparent decrease in diversity during the warm season (Fig. 3) might be interpreted as an effect of increasing the abundance of fast growing microorganisms—e.g., greater abundance of *DA101* during the warm season (Table 2), rather than the loss of microbial species. The concept that observed changes in community composition are actually variations in the relative abundance of taxa rather than extinction and recolonization of taxa in the ecosystem was previously raised by *Caporaso et al. (2012b)* in a study of seasonal dynamics in microbial community composition of the Western English Channel. Here, most microbial taxa are always present in these soils but vary in abundance with shifts in seasons. To validate this hypothesis a deeper sequence survey should be performed within our samples in order to obtain a better picture of the rare biosphere (*Lynch & Neufeld, 2015*).

Despite the similar number of observed species found between seasons, samples collected during the warm season were more heterogeneous and presented greater variability than those collected in the cold season (Fig. 3). Cold temperatures allowed for greater evenness while warm temperatures caused an apparent decrease in diversity. In the cold temperatures, a more even abundance distribution allows more taxa to be sampled when the same number of sequences is examined, causing the richness to appear higher. Temperature variation restricts survival of a few species or genera sensitive to a specific temperature condition (*Li et al., 2015*), which corroborate the aforementioned statement. As the amount of rain was well distributed along the year (Fig. 1), the temperature was considered the main source of environmental variation for microbial life in our experiment.

Temperature may directly affect microbial metabolism, by restricting the activity of those microbes unable to thrive at low temperatures. Temperature can also indirectly affect microbial communities by reducing plant growth, thereby decreasing carbon rhizodeposition (*Murphy, Foster & Gao, 2016*). Most plant species of subtropical grasslands are perennial but present lower growth rate and higher senescence rate during the winter.

Thus, rhizodeposition of available carbon sources is reduced during the winter which in turn affects fast growing microbial populations. Furthermore, fast-growing microbial populations might be adapted to easily degradable materials during the winter due to the plant litter decomposition while slow-growing microorganisms are favored when substrate availability is limiting (*Rui, Peng & Lu, 2009*). Thus, regardless of the environmental conditions (warm or cold), the different soil niches will be likely to be occupied by some member of the microbial community. The level of environmental constraints imposed by the subtropical region on bacterial communities is significant, encouraging functional redundancy and resilience to be further explored.

Although the soil, vegetation cover, temperature, and rainfall regime are similar in the Pampa biome and the sampling areas were relatively close to each other, the soil microbial community structure was also affected by location (Fig. 2). These differences are related to the inherently high level of spatial heterogeneity observed in soil (*Lauber et al., 2013*). While we attempted to sample areas as homogeneous as possible, and besides the similarity between soils from the two areas, the natural variation between soils resulted in differences between soils of 77%, 26%, 73%, 67%, 108% and 133% in the contents of Al, P, Ca, Mg, Zn e B, respectively. Besides, as mentioned above, vegetation cover was similar but no identical. The species *Aristida laevis* and *Elephantopus mollis* were detected only in SM while the plant species *Axonopus affinis*, *Eragrostis plana*, *Erianthus angustifolius*, *Paspalum dilatatum*, *Sporobolus indicus* and *Vernonia nudiflora* were only detected in SG. Finally, the presence of the highly invasive and allelopathic plant, *Eragrostis plana*, can alter important ecological components of microhabitats (*Guido et al., 2016*). Therefore, these differences in soil and vegetation may result in different abiotic conditions between the two sites, capable of altering the composition and activity of the soil microbial community.

## CONCLUSIONS

Here we reported the effects of seasons and location on soil microbial taxonomy and functional profiles providing insights in the ecological rules shaping soil microbial communities and their functions in natural ecosystems. Season and location significantly modulate microbial community assemblage and abundance in subtropical natural grasslands. However, our data suggest that grasslands maintained a stable microbial community membership along the year with oscillation in abundance. Apparently soil microbial taxa are more susceptible to natural climatic disturbances while functions are more stable and change with less intensity along the year. Finally, our data allow us to conclude that the most abundant microbial groups and functions were shared between seasons and locations reflecting the existence of a stable taxonomical and functional core microbiota.

## ACKNOWLEDGEMENTS

The authors acknowledge Dra. Manoeli Lupatini for taking soil samples from the Santa Maria site. We also like to thank the Fundação Estadual a Pesquisa Agropecuária do Estado

do Rio Grande do Sul (FEPAGRO) for providing the experimental area for sampling in São Gabriel.

### Funding

This study was funded by the Conselho Nacional de Desenvolvimento Científico e Tecnológico (CNPq), Fundação de Amparo à Pesquisa do Rio Grande do Sul (FAPERGS) and by the Brazilian Microbiome Project (http://www.brmicrobiome.org). Victor S. Pylro received a fellowship from FAPESP (Process number 2016/02219-8). The funders had no role in study design, data collection and analysis, decision to publish, or preparation of the manuscript.

### Grant Disclosures

The following grant information was disclosed by the authors:
Conselho Nacional de Desenvolvimento Científico e Tecnológico (CNPq).
Fundação de Amparo à Pesquisa do Rio Grande do Sul (FAPERGS).
Brazilian Microbiome Project.
FAPESP: 2016/02219-8.

### Competing Interests

The authors declare there are no competing interests.

### Author Contributions

- Anthony Diego Muller Barboza and Victor Satler Pylro conceived and designed the experiments, performed the experiments, analyzed the data, prepared figures and/or tables, authored or reviewed drafts of the paper, approved the final draft.
- Rodrigo Josemar Seminot Jacques, Paulo Ivonir Gubiani and Júlio Kuhn da Trindade conceived and designed the experiments, performed the experiments, contributed reagents/materials/analysis tools, authored or reviewed drafts of the paper, approved the final draft.
- Fernando Luiz Ferreira de Quadros conceived and designed the experiments, contributed reagents/materials/analysis tools, authored or reviewed drafts of the paper.
- Eric W. Triplett conceived and designed the experiments, performed the experiments, authored or reviewed drafts of the paper, approved the final draft.
- Luiz Roesch conceived and designed the experiments, performed the experiments, analyzed the data, contributed reagents/materials/analysis tools, prepared figures and/or tables, authored or reviewed drafts of the paper, approved the final draft.

### DNA Deposition

The following information was supplied regarding the deposition of DNA sequences:
Raw sequences obtained by metatranscriptome sequencing and associated metadata were submitted to MG-RAST server (http://metagenomics.anl.gov/) and are publicly available under the string mgp12046. All raw sequences obtained by amplicon sequencing were

submitted to NCBI Sequence Read Archive (SRA) and are available under the experiment number SRX2779549 and run number SRR5499445.

## Data Availability

The raw data is included in the Bioinformatics analysis and statistics section.

## Supplemental Information

Supplemental information for this article can be found online at http://dx.doi.org/10.7717/peerj.4991#supplemental-information.

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
