# Peer review of "Seasonal dynamics alter taxonomical and functional microbial profiles in Pampa biome soils under natural grasslands"

_PeerJ, doi:10.7717/peerj.4991_

## Round 0.1 · original submission · Major Revisions

Both reviewers and myself saw your manuscript of interest. However there are some issues that need to be revised and several comments taken into account. Please, address carefully especially those related to statistical analyses or interpretation of the results.

Reviewer 1 ·

Basic reporting

Please, see General Comment

Experimental design

Please, see General Comment

Validity of the findings

Please, see General Comment

Additional comments

This is a noteworthy, sound and welcome contribution to the field of soil microbial ecology. It is well conceived and designed, uses modern metagenomic methodologies, results are very ably analysed, and pertinent conclusions are drawn.
In my view, three general conclusions stand out:
1) the soil microbiota functional and taxonomic composition responds to seasonal and location variations;
2) Abundance of specific taxons is more sensitive to environmental change than functional capabilities of the microbiota;
3) a core fraction of the microbiota, that probably defines the specific biome being studied, persists over time and space.

The fact that the ms. does not need to go in-depth into specificities of the taxonomic and functional characterisation attests to the relevance of this study.

Among the strong points in the manuscript are the use of metatranscriptomics to analyse the functional composition of soil microbiota, an adequate level of replication, and a thorough statistical treatment of the significance of the data obtained.

Despite the overall, highly positive impression, I must note some specific points that, in my opinion, would lead to an improvement of the ms. if taken into consideration:

1. The abstract is written in a rather obscure and convoluted manner, and it is difficult to understand. I believe it would strongly benefit from a rewriting along the lines of what is more clearly stated in the Conclusions section.

2. In the introductory section, when authors introduce available scholarship on the effect of environmental factors on soil microbiota, they omit pH, a major driver of microbial diversity in the soil. I believe some of the current scholarship on that factor should be mentioned.

3. Throughout the ms. authors run into an apparent contradiction. They claim to expect a higher abundance and diversity in warm seasons (l. 87), and when they get quite the opposite (l. 244-248), they rationalize it by stating that this is due to an increased abundance of fast growing microorganisms in summer (l. 330-338) without disappearance of other taxons. If this were indeed the case, I believe the observed results would be a direct consequence of a not-deeep-enough sequence coverage, and authors should elaborate on that.

4. I was a little surprised that authors chose to use functional metatranscriptomics, a decision to be applauded, but chose to amplify 16S rDNA rather than 16S rRNA, which would have allowed them to infer the metataxonomics of functional microorganisms.

5. I may have missed it, but, did authors compare metataxonomic assignments derived from amplicon metagenomics with those derived from metatranscriptomics? This would enrich the ms.

6 . Discussion on growth on nucleic acids (L. 323-329) is probably irrelevant, based on a reference (Tozzi et al) that is probably not appropriate, and could probably be left out.

·

Basic reporting

In this manuscript, Barboza et al. have used Next Generation Sequencing to compare the taxonomic (metataxonomics) and functional (met transcriptomics) profiles among soil samples from two subtropical natural grasslands located in the Pampa biome (Brazil) considering location and marked seasonal environment variation. They reported effects of season on both microbial community structure (abundance and richness) and functions. Also, the authors state that the seasonal variation mainly come from oscillation in the relative abundances of specific taxa along seasons. Finally argued the existence of a stable taxonomical and functional core microbiota associated to the biome.

The article is well set and described in all its parts (with few exceptions pointed below). The results and discussion section are accurately written and takes into account the data and justifies them. This kind of research increase our perception about microbial community ecology and physiology dynamic interaction with environmental factors, being a step forward to build the complex operational concept of soil healthy.

Experimental design

a. The experimental setup cover two location (SM and SG), two seasons (cold and warm) with 3 replicates. For the same season, there are two samples (may and august -cold) and (nov and feb - warm). Why there is no data comparing possible changes in microbial phylotypes and/or microbial functions? It would represent a knowing variation for local effect. We see no data in the article for this. why?

Validity of the findings

I would like to raise some points related to the validity of the findings:

a. The authors properly claimed that environmental factors such as temperature and moisture are main drivers of microbial community assembly. And the former is decisive to explain the experimental finds. Environmental factors are assumed to shape structural and functional diversity of microbial communities living in soil. However, for soils under influence of plant root community (rhizosphere effect), a new drive force emerges. Do the results obtained in this study coming from soils free of root exudates of the plant community? If not, the data interpretation related to location and seasonal effect will be strongly biased.

b. PCA graphic representation had shown clear distinction for phylotypes in location and season. Should be interesting highlight the contrasting pattern for presence/absence and abundance related to microbial functions. (Pg. 10, lines 3-5)

c. For PERMANOVA analysis (table 1) my first point is that for the microbial phylotype parameters, location and season interaction was significant at 5% probability. It means that we cannot infer the data for location and season separated. The community structure shifts emerge from the interaction factors. I think it has an impact in the data interpretation. For microbial function, there is no interaction and even location effect, so the evaluated microbial functionalities were preserved despite of the changes in microbial community. In this case, seasonal effects have a role, being the quantitative effect more pronounced than the qualitative. For functionality, it would be possible combine two location to strength the interpretation of season. Please show your view about the comment raised.

d. Even considering the same biome, climate and soil taxonomy for soil sampling, soil microscale variation will be an important factor, so it is no possible to avoid location effect over microbial structure as stated in pg. 10 (lines 9-14).

e. Why the cut-off of relative frequency greater than 1%? How representative is the sum of the others below 1%? We could group them in highest taxonomy order and bring them to be analyzed?

f. Table 2 show the major groups. Why the results were expressed sometimes for species, genera, or families? Why the diversity of predominant groups is low? The establishment of a core in this situation could be critic. Why do not consider the group of less frequency to compose the core?

g. Pg. 11 - lines 8 - 11: The authors mentioned that" In agreement with the differential abundance analysis, alpha diversity measurements indicated similar microbial richness (number of observed OTUs) during the seasons, but greater microbial diversity during the cold season (Fig. 3)". My comment: It is not so clear for Shannon index applied to taxa diversity.

h. Pg. 11 (lines 10--12): "Cold temperatures allowed for greater evenness while warm temperatures decrease the diversity of taxa" Why we cannot see it for observed alpha diversity? (Fig 3)

i. Pg. 11 -lines 12-13: The statement "The diversity of microbial functions presented similar trends, displaying a core set of microbial functions along the year in both locations (Fig. 3 and Table 3)". My comment: What you mean similar trend in this context?

Additional comments

Some minor points:
a. Soil samples were collected under natural grassland, currently used for grazing of cattle. Question: Is the sampled soil free of plant influence?

b. Please check the statement (pg. 5 - lines 14 -17) and compare with the raw data fig 1 plus fig 1. for example, rainfall regime for February 15th (check averages)

c. Pg. 5 - Lines 18 to 22: What is the relevance of this information for the study?

d. Note that PDF article submitted version fig 2 and fig 3 were misplaced (referred at pg. 10, line 2-3)

e. Figure 3 Legend seems to be incomplete

f. Pg. 11 (line 22-23): Legend Table 3. Is it correct? “Number highlighted in bold represent greater abundance during either cold or warm season.”

g. Pg. 12-13. line 24 and following: Explain why? "Our sampling strategy allowed us to obtain a highly tractable model, with true landscape level of biological replications, for verifying whether soil microbial functions follow the changes in the abundance of taxa during seasonal variations"

---

## Round 0.2 · accepted · Accept

All the comments and criticism raised during the review process have been satisfactorily solved.

# Reviewer 1 ·

Basic reporting

My assessment of the previous version of this manuscript was, overall, highly positive. This version has taken into consideration the reviewers' suggestions and has clarified some of the unclear points mentioned in my previous review, making it a better, more balanced contribution.

Experimental design

Some of the doubts arising out of the previous version have been satisfactorily solved. No further comment.

Validity of the findings

Again, this is a well rounded manuscript reporting on relevant observations, well carried out, analysed and discussed. Overall, a worthwhile addition to soil microbiomics

·

Basic reporting

The authors covered properly criticisms and question raised. As anticipated in my first review the contribution is scientifically sound and deserve a place on Peer J.

Experimental design

After R1-version of the manuscript, I have no additional comment related to experimental design.

Validity of the findings

After R1-version of the manuscript, I have no additional comment related to validity of the findings.

Additional comments

No additional comments